

# ConFindr: rapid detection of intraspecies and cross-species contamination in bacterial whole-genome sequence data

Andrew J. Low[*], Adam G. Koziol, Paul A. Manninger, Burton Blais and Catherine D. Carrillo[*]

Ottawa Laboratory (Carling), Canadian Food Inspection Agency, Ottawa, Ontario, Canada
[*] These authors contributed equally to this work.

## ABSTRACT

Whole-genome sequencing (WGS) of bacterial pathogens is currently widely used to support public-health investigations. The ability to assess WGS data quality is critical to underpin the reliability of downstream analyses. Sequence contamination is a quality issue that could potentially impact WGS-based findings; however, existing tools do not readily identify contamination from closely-related organisms. To address this gap, we have developed a computational pipeline, ConFindr, for detection of intraspecies contamination. ConFindr determines the presence of contaminating sequences based on the identification of multiple alleles of core, single-copy, ribosomal-protein genes in raw sequencing reads. The performance of this tool was assessed using simulated and lab-generated Illumina short-read WGS data with varying levels of contamination (0–20% of reads) and varying genetic distance between the designated target and contaminant strains. Intraspecies and cross-species contamination was reliably detected in datasets containing 5% or more reads from a second, unrelated strain. ConFindr detected intraspecies contamination with higher sensitivity than existing tools, while also being able to automatically detect cross-species contamination with similar sensitivity. The implementation of ConFindr in quality-control pipelines will help to improve the reliability of WGS databases as well as the accuracy of downstream analyses. ConFindr is written in Python, and is freely available under the MIT License at github.com/OLC-Bioinformatics/ConFindr.

## INTRODUCTION

Public-health microbiology laboratories increasingly apply bacterial whole-genome sequence (WGS) analyses for pathogen identification, high-resolution typing and risk profiling (*Ronholm et al., 2016*; *Allard et al., 2016*; *Taboada et al., 2017*). Reductions in cost for generating WGS data have led to the widespread use of this technology for tracking foodborne pathogens internationally, and public databases currently include sequences for hundreds of thousands of isolates. A significant effort has been undertaken to produce guidelines and minimum standards for sequence data quality, particularly when such

Corresponding author
Catherine D. Carrillo,
catherine.carrillo@canada.ca

data is used to support regulatory activities (*Lambert et al., 2017*; *Rossen, Friedrich & Moran-Gilad, 2017*).

Quality assessment tools are typically integrated into bioinformatics workflows to ensure the reliability of WGS data (*Koren et al., 2014*; *Page et al., 2016*). For example, FastQC is used to assess the per-base quality of raw reads to identify problems with the sequencing libraries or runs (*Andrews, 2010*). Tools such as QUAST can be used to evaluate the quality of *de novo* assemblies, identify misassemblies, determine error rates, and more (*Gurevich et al., 2013*). These tools can be extremely valuable for identifying inferior datasets; however, assessing contamination is outside of their current scope.

The presence of contamination in WGS data is recognized as an important sequence quality issue (*Merchant, Wood & Salzberg, 2014*; *Ballenghien, Faivre & Galtier, 2017*; *Robertson et al., 2018*; *Cornet et al., 2018*; *Flickinger et al., 2015*). Introduction of contaminants can occur at many stages in the generation of bacterial sequence data. For example, cultures recovered from samples may not be adequately purified, or cross-contamination could occur during preparation of genomic DNA or sequencing-libraries (*Merchant, E Wood & Salzberg, 2014*). Carryover contamination results from the presence of residual fragments from previous sequencing runs (*Souvorov, Agarwala & Lipman, 2018*). While integration of controls can help to identify pervasive contamination issues, they are not effective for the identification of sporadic contamination events.

Cross-species contamination in short-read WGS data can be readily identified by taxonomic classification of sequence reads using reference databases  (*Wood & Salzberg, 2014*; *Merchant, E Wood & Salzberg, 2014*; *Ounit et al., 2015*; *Mallet et al., 2017*). Contamination can also be inferred following *de novo* assembly of short sequencing reads into a contiguous bacterial chromosome. Contiguity can be impacted by presence of contaminating sequencing reads, but also by factors such as the assembler used, length of the sequencing reads, presence of repeat regions, GC content and coverage (*Lin et al., 2011*; *Jünemann et al., 2014*; *Souvorov, Agarwala & Lipman, 2018*). Contamination may be indicated by a highly fragmented assembly, or a genome size that is larger than expected (*Robertson et al., 2018*). However, establishment of appropriate cutoffs requires determination of acceptable ranges within a species, and atypical strains may fall outside of these limits.

Intraspecies contamination is far more difficult to detect because read-classification approaches cannot be used. Effective tools have been developed for detection of contamination in human genome studies (*Flickinger et al., 2015*), but these approaches are not easily transferable to bacterial samples due to high variability in bacterial genome content, even within a species. In some studies, the quality of metagenomic or single-cell sequencing data is assessed by evaluating core genes to determine the completeness and degree of contamination of assemblies (*Hess et al., 2011*; *Parks et al., 2015*). One of the tools developed for this purpose is CheckM, which determines the presence of contamination based on the identification of multiple copies of lineage-specific, ubiquitous, single-copy genes (*Parks et al., 2015*). To our knowledge, there are no tools designed and evaluated specifically for the detection of intraspecies contamination in bacterial-isolate sequence data.

We have developed a bioinformatics tool, ConFindr, which can accurately and rapidly identify intra- and cross-species contamination based on the analysis of raw sequencing reads. We evaluated the performance of this tool for detecting contamination in Illumina short-read WGS data derived from priority foodborne pathogens *Listeria monocytogenes*, *Salmonella enterica* and Shiga-toxin producing *Escherichia coli* (STEC).

## METHODS

### ConFindr workflow and implementation

ConFindr determines the presence of contaminating sequencing reads based on the analysis of the set of 53 genes encoding the bacterial ribosomal-protein subunits that are used in the ribosomal multilocus sequence typing scheme (rMLST) (*Jolley et al., 2012*). The rMLST genes are typically present as single copies and are conserved across the entire bacterial domain, with some exceptions where multiple alleles for a gene exist or no gene exists. ConFindr works on the principle that a genome containing more than one allele for any rMLST gene is contaminated, taking into consideration the known exceptions.

In its first step, ConFindr uses a screening functionality provided in Mash (*Ondov et al., 2016*) to determine which genera are present in a sample. This screen is done against a custom database derived from the NCBI RefSeq genomes (https://www.ncbi.nlm.nih.gov/refseq/) with one genome representing each species (*O'Leary et al., 2015*). If more than one genus is detected, ConFindr reports cross-species contamination for the sample and does not proceed further. If only one genus is present, ConFindr creates a genus-specific rMLST database by extracting all rMLST sequences associated with the target genus, excluding genes known to have multiple alleles, and proceeds to attempt to find contamination by searching for multiple alleles of one or more of the benchmark rMLST genes.

To search for multiple alleles, ConFindr begins by using BBDuk (*Bushnell, 2014*) to extract reads that are likely part of the rMLST gene set. These baited reads are stringently trimmed, again using BBDuk, and then aligned to the rMLST genes using BBMap (*Bushnell, 2014*). The resulting BAM file is then parsed in order to find 'Contaminating Single Nucleotide Variants' (cSNVs)—that is, sites in the pileup where more than one base is present. Since all rMLST genes are known to be present as single copies, the occurrence of multi-base sites in the pileup indicates multiple alleles, and therefore contamination. To be called as a cSNV, at least 2 bases of the minor variant with a Phred score of 20 or greater must be present at that site, and at least 5 percent of bases must support the minor variant (though these parameters can be changed by the user). ConFindr determines that a sample is contaminated if multiple genera are found in the Mash screen step or if three or more cSNVs are found.

### In silico dataset creation

To create *in silico* datasets, we selected complete assemblies from RefSeq for *E. coli*, *S. enterica*, and *L. monocytogenes* (accessions NC_002695.1, NC_003198.1, and NC_003210.1, respectively) and generated variants of these genomes with 100, 500, 1,000, and 2,000 SNVs using a custom script available at https://github.com/lowandrew/MutantCreator. Simulated reads were then created from both variant and base genomes using ART v2.5.8

**Table 1  Summary of test dataset used for ConFindr Evaluation.**

| Target strain | Contaminant strain | rMLST SNVs | ANI | Contamination levels |
|---|---|---|---|---|
| E. coli O121:H19 (OLF17053-3) | E. coli O121:H19 (OLC2152) | 0 | 99.98 | 0, 20 |
| | E. coli O121:H19 (OLC2152) | 11 | 98.86 | 0, 5, 10, 20 |
| | E. coli O15:H14 (OLF17030) | 11 | 98.78 | 0, 5, 10, 20 |
| | E. coli O8:H28 (OLF17043) | 11 | 98.83 | 0, 5, 10, 20 |
| | Enterobacter cancerogenus (OLC1687) | N/A | 78.86 | 0, 5 |
| S. Heidelberg (OLC2542) | S. Heidelberg (OLC2000) | 0 | 99.99 | 0, 20 |
| | S. Bredeney (OLC2229) | 32 | 98.36 | 0, 5, 10 20 |
| | S. Typhimurium (OLF13104-7) | 24 | 99.08 | 0, 5, 10 20 |
| | S. Dublin (OLF18064-1) | 33 | 98.82 | 0, 5, 10 20 |
| | Citrobacter freundii (OLC1136) | N/A | 81.83 | 0, 5, 10 20 |
| L. monocytogenes (OLF10129) | L. monocytogenes (OLF11041-1) | 0 | 99.99 | 0, 20 |
| | L. monocytogenes (OLF13043-2) | 16 | 99.54 | 0, 5, 10, 20 |
| | L. monocytogenes (OLF15140) | 10 | 99.45 | 0, 5, 10, 20 |
| | L. monocytogenes (OLF09168) | 133 | 94.85 | 0, 5, 10, 20 |
| | Listeria innocua (OLC0004) | 420 | 88.26 | 0, 5, 10 |
| | Enterococcus faecalis (OLC0147) | N/A | 66.36 | 0, 5 |

(commands used to carry this out can be found in Table S2) (*Huang et al., 2012*) and mixed together in proportions of 0, 1, 5, 10, and 20 percent contamination using scripts found at https://github.com/lowandrew/FastQMixer to a total coverage depth of approximately 60X. Five replicates were created for each mixed read set.

## Test datasets

We generated a test dataset of 48 samples comprised of intra- and cross-species mixes of *E. coli*, *S. enterica*, and *L. monocytogenes* isolates with varying levels of relatedness (Table 1). Average nucleotide identity (ANI) was calculated using OrthoANI version 1.4.0 (*Lee et al., 2016*). This dataset was made both *in silico* by mixing together reads from previous runs of these isolates using the reformat.sh program of the BBMap package (*Bushnell, 2014*) to a coverage depth of 80X, as well as by sequencing lab-generated mixes of genomic DNA (gDNA).

To generate WGS data, bacterial isolates were cultured in Brain Heart Infusion (BHI) broth (Oxoid Ltd., Basingstoke, Hampshire, England) for 4 to 6 h at 36 °C, and gDNA was extracted using the Maxwell 16 Cell SEV DNA Purification Kit (Promega, Madison, WI). DNA was quantified using the Quant-it High-Sensitivity DNA Assay Kit (Life Technologies Inc., Burlington, ON). Sequencing libraries were constructed from 1 ng of gDNA using the Nextera XT DNA Sample Preparation Kit (Illumina, Inc., San Diego, CA) and the Nextera XT Index Kit (Illumina, Inc.) according to manufacturers' instructions. Genomic sequencing was performed on the Illumina MiSeq Platform (Illumina, Inc.) using a 600-cycle MiSeq Reagent kit v3 (Illumina, Inc.).

## Nucleotide sequence accession numbers

Raw data have been deposited at DDBJ/EMBL/GenBank under BioProject PRJNA507762. The accession numbers and strain descriptions are listed in Table S1.

## Coverage dataset

To evaluate the effect of extremely high coverage depth as well as different sequencing platforms on the performance of ConFindr, we created simulated datasets from RefSeq assemblies (see Table S5) for both the HiSeq and MiSeq platforms using ART v2.5.8 (commands in Table S2) at coverage levels of 50, 100, 200, 300, 400 and 500×, and at contamination levels of 0, 5, 10, and 20 percent.

## Genome assemblies and quality metrics

All of the read sets created were also put through the process of *de novo* assembly. Briefly, reads were quality trimmed using bbduk.sh and error corrected with tadpole.sh (both of the BBMap package (*Bushnell, 2014*)) and then assembled using SKESA v2.3.0 (*Souvorov, Agarwala & Lipman, 2018*). Exact commands used to carry this out can be found in Table S2. Genome quality statistics were assessed using QUAST v4.6.3 (*Gurevich et al., 2013*).

## Calculation of number of SNVs between rMLST types

To calculate the number of SNVs between rMLST types within *E. coli*, *S. enterica*, and *L. monocytogenes*, we retrieved all rMLST allele sequences and the list of profiles (accessed at https://pubmlst.org/rmlst/, November 1, 2018). We then extracted sequences for each allele within each rMLST type (1,641 types for *L. monocytogenes*, 3,062 types for *E. coli*, and 7,255 for *S. enterica*). The number of SNVs between every sequence type pair within each species was calculated by aligning each gene in the first type against each gene in the second type using the pairwise2 module in biopython (*Cock et al., 2009*).

## Dataset testing

To detect contamination in the datasets generated, ConFindr v0.4.4 was run on default settings on all samples generated. Kraken v1.0 (*Wood & Salzberg, 2014*) was run on fastq files that had been trimmed to a quality of 15 with bbduk.sh against the standard Kraken database. Exact commands used to carry this out can be found in Table S2. Strains/species were determined to be present if at least 0.5 percent of classified reads could be assigned to them. CheckM v1.0.11 (*Parks et al., 2015*) was run with the lineage_wf workflow for each assembly, and samples were called as contaminated if their contamination level was 0.6 percent or greater, as across our test dataset our uncontaminated samples had up to 0.6 percent contamination called by CheckM.

# RESULTS

## Identification of contaminating SNVs within rMLST genes using ConFindr

As ConFindr is based on finding contaminating SNVs (cSNVs) within the rMLST genes in raw reads, we first looked at the reliability of detection of cSNVs in simulated data

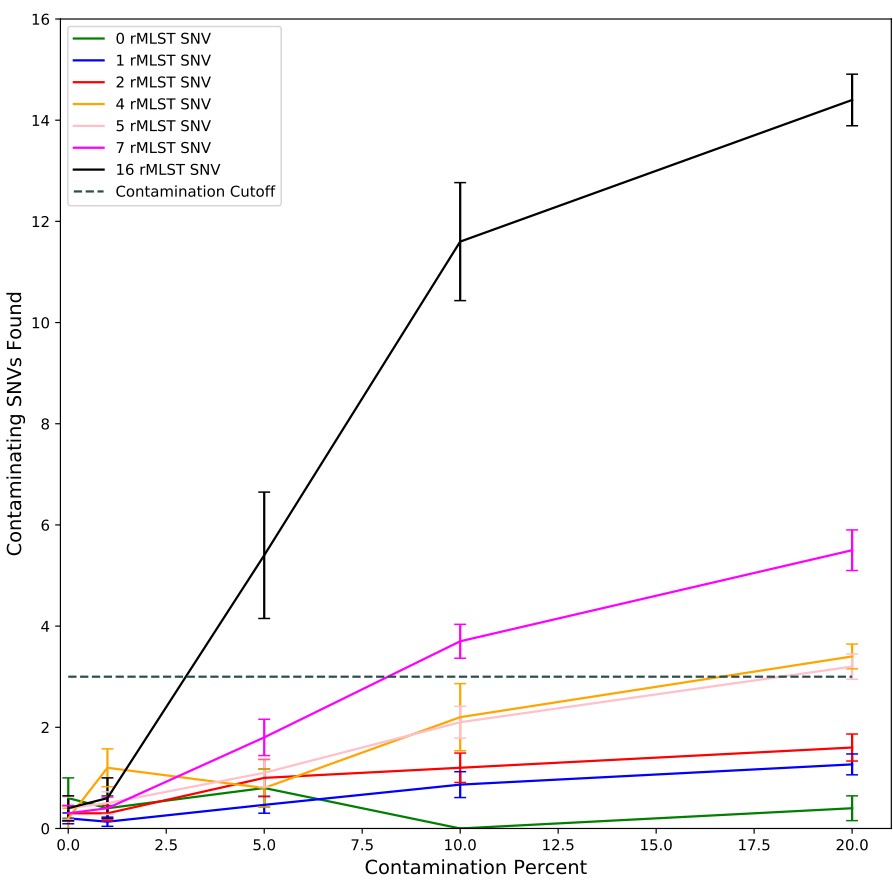

**Figure 1** **Detection of contaminating SNVs within rMLST genes by ConFindr.** Sequencing reads were generated *in silico* from complete assemblies for *E. coli*, *S. enterica*, and *L. monocytogenes* and synthetic mutants containing 100, 500, 1,000 and 2,000 randomly-distributed SNVs. Reads were mixed to generate datasets with 0, 1, 5, 10 and 20% contamination. Datasets were binned according to the number of SNVs (0 to 16) occurring within rMLST genes in the contaminant relative to the parent strain (Table S3). The number of contaminating SNVs identified in each dataset was plotted relative to percent contamination of the sample. Error bars indicate standard error with 5 replicates for 0, 4, 5, and 16 SNVs, 10 replicates for 2 and 7 SNVs, and 15 replicates for 1 SNV.

with different levels of contamination. Synthetic mutants with 100 to 2000 SNVs relative to reference genomes were generated *in silico*, and the number of SNVs occurring within rMLST genes in these mutants was calculated. The number of cSNVs found by ConFindr in the *in silico* datasets at 60 times coverage was compared with the predicted number of SNVs within rMLST genes in the two isolates making up the contaminated sample (Fig. 1). As the relative contamination increased, ConFindr's estimate of the number of cSNVs in the sample approached the expected number of SNVs. Contamination at 5% was reliably detected when the contaminant had at least 16 SNVs within the rMLST genes, at 10% with at least 7 SNVs and at 20% with at least 3 SNVs within target genes.
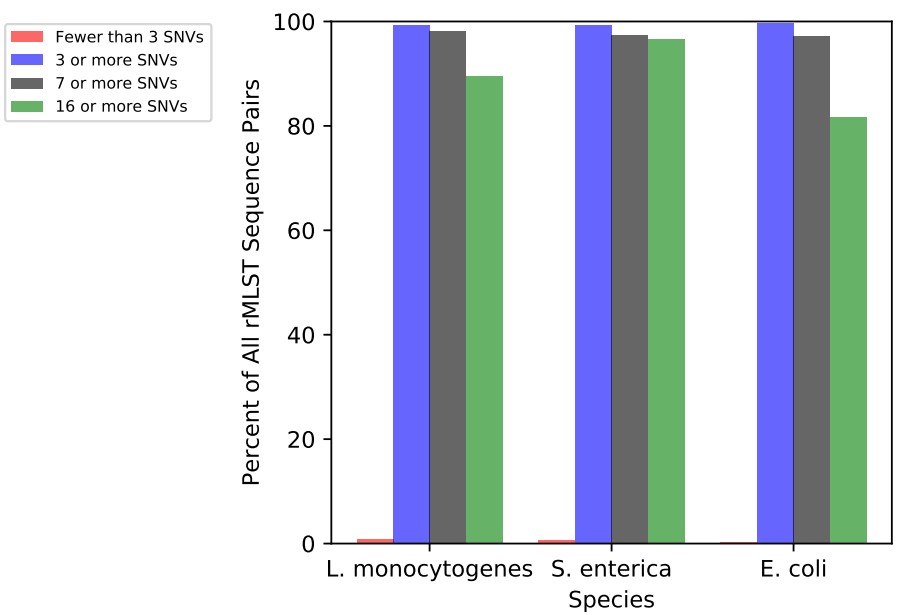

**Figure 2** SNV distance between all pairs of rMLST sequence types for *L. monocytogenes*, *S. enterica*, and *E. coli*.

## Diversity among rMLST sequences types

To illustrate the genetic diversity within the rMLST scheme, we calculated the numbers of SNVs between ribosomal sequence types (rSTs) for *L. monocytogenes*, *S. enterica*, and *E. coli* (Fig. 2). Over 99 percent of all pairs in all three species had three or more SNVs, which is the cutoff chosen in ConFindr as the minimum number of cSNVs that need to be found before a sample will be considered contaminated. Over 80 percent of the sequence types have 16 or more SNVs relative to others. Therefore, ConFindr should almost always be able to detect contamination between two isolates with different rMLST types.

## ConFindr detects contamination with more sensitivity that existing tools

We compared ConFindr to existing tools capable of detecting contamination, CheckM and Kraken using both *in silico* and lab-generated datasets. Mixes were binned based on the Average Nucleotide Identity (ANI) of the two samples being mixed - those with >99 percent ANI, representing very closely related mixes, those with between 98 and 99 percent ANI, representing same-species mixes between strains not as closely related, and those with ANI of less than 98 percent, representing distantly related same-species mixes and cross-species mixes (Table 1).

ConFindr was more sensitive than either CheckM or Kraken for intraspecies contamination detection (Figs. 3A, 3B, 3D, and 3E), and comparable to both for cross-species contamination detection (Figs. 3C and 3F). ConFindr detected contamination successfully in all cases except for one simulated mix of closely related strains at 5 percent

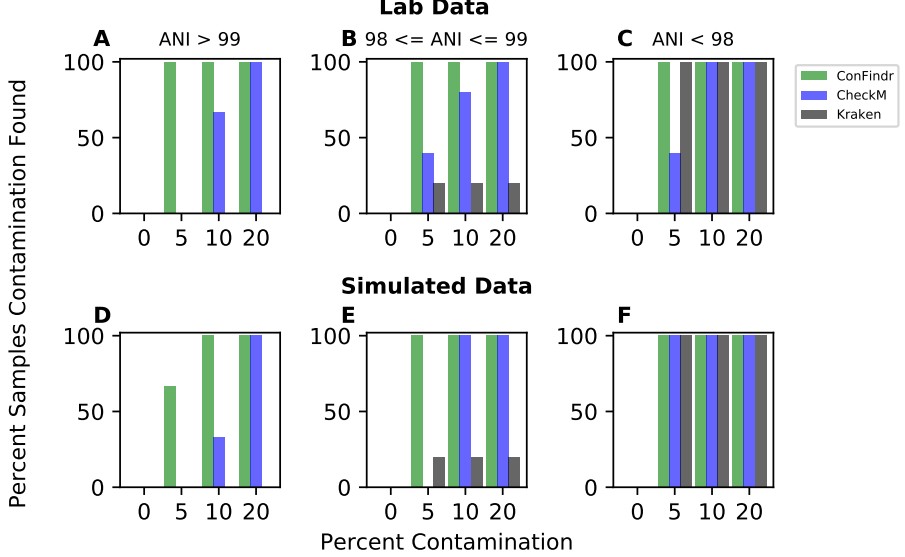

**Figure 3** **Performance of ConFindr compared with CheckM and Kraken for closely related, related, and distantly related mixes (ANI > 99, between 98 and 99, and <98, respectively).** A, B, and C show results using laboratory data, while D, E, and F show results with simulated data.

contamination (Table S1), while both CheckM and Kraken required either more distance between species or a higher level of contamination for its determination.

To further verify improved sensitivity of ConFindr relative to CheckM or Kraken, we varied the cutoffs used for each approach. Across our test datasets, the contamination percentage reported by CheckM for uncontaminated samples varied from 0 to 0.6 percent. As shown in Fig. 3, at this cutoff CheckM was not as sensitive as ConFindr. Lowering this cutoff further in an attempt to achieve increased sensitivity resulted in a proliferation of false positives, while still not achieving sensitivity equivalent to ConFindr. For example, lowering the cutoff to 0.3 resulted in 13 of 72 contaminated samples being falsely labelled as clean, with 5 of 16 clean samples being falsely labelled as contaminated. Further lowering the cutoff to classify anything with a reported contamination greater than 0 as contaminated resulted in 7 of 72 contaminated samples being labelled as clean, and 10 of 16 clean samples labelled as contaminated. Similar results were found for Kraken—lowering the cutoff used to a value of 0.05 resulted in false negative calls for 12 of 72 contaminated samples, while false positive results are reported for 10 of 16 clean samples. For the same dataset, ConFindr has only 1 false negative, and 0 false positives.

## Assembly metrics are insufficient for contamination detection

We assembled the contaminated datasets used in this study to assess metrics such as number of contigs, N50 and total length for contaminated datasets relative to uncontaminated datasets. At 5% contamination, there was an increase in the number of contigs and the total length of the assembly, and a decrease in N50 (Table 2). The relative increase or decrease in these metrics varied depending on the strain used as the contaminant. For example, contamination of *L. monocytogenes* strain OLF10129 with OLF15140 appeared to

**Table 2** Assembly metrics for intraspecies contamination dataset.

| Strain 1 | Strain 2 (contaminant) | Percent contaminant | N50 | # Contigs | Total length |
|---|---|---|---|---|---|
| *L. monocytogenes* (OLF10129) | NA | 0 | 338684 | 16 | 2966006 |
| | OLF13043-2 | 5 | 291474 | 47 | 3007464 |
| | | 10 | 112978 | 83 | 3073200 |
| | OLF15140 | 5 | 302686 | 26 | 2971529 |
| | | 10 | 134503 | 78 | 3008481 |
| | OLF09168 | 5 | 320971 | 57 | 2992774 |
| | | 10 | 271165 | 453 | 3333771 |
| *S. Heidelberg* (OLC2542) | NA | 0 | 693768 | 29 | 4856249 |
| | *S. Bredeney* (OLC2229) | 5 | 381278 | 34 | 4859111 |
| | | 10 | 228154 | 111 | 4936326 |
| | *S. Typhimurium* (OLF13104-7) | 5 | 235407 | 66 | 4910684 |
| | | 10 | 162612 | 162 | 5080654 |
| | *S. Dublin* (OLF18064-1) | 5 | 387491 | 56 | 4881296 |
| | | 10 | 272678 | 132 | 5022778 |
| *E. coli* O121:H19 (OLF17053-3) | NA | 0 | 134602 | 196 | 5150254 |
| | O174:H19 (OLF17021-7) | 5 | 122561 | 234 | 5183143 |
| | | 10 | 50560 | 433 | 5337747 |
| | O8:H28 (OLF17043) | 5 | 119304 | 197 | 5154985 |
| | | 10 | 31707 | 477 | 5329336 |
| | O15:H14 (OLF17030) | 5 | 121490 | 230 | 5179176 |
| | | 10 | 32121 | 509 | 5385110 |

have a smaller impact than contamination with the more distantly related isolate OLF09168. Statistics on all assemblies at all contamination levels are available in Table S1.

## Effect of sequencing coverage and platform

To verify that ConFindr does not produce false positives at high sequencing depth and works across both the HiSeq and MiSeq platforms, we created a simulated data set from RefSeq genomes at coverage levels of up to 500X. ConFindr found no false positives on any uncontaminated samples, and only failed to detect contamination in one closely related *L. monocytogenes* mix (50X HiSeq coverage of NC_002973 and NC_012488 at 5 percent contamination). ConFindr also did not call mixes between two closely related *E. coli* samples (NC_002695 and NC_011353) as contaminated, but this was due to the fact that the rMLST genes between these two samples contained only 2 SNVs, below the threshold of 3 SNVs that ConFindr uses to call contamination—the 2 SNVs present were accurately detected at many of the coverage and contamination levels tested. The full results for this dataset are available in Table S5.

## Contamination in SRA data

To evaluate the prevalence of contamination in public databases, we randomly selected 1,500 isolates sequenced on Illumina instruments from the Sequence Read Archive (SRA) —500 for each of *E. coli*, *S. enterica*, and *L. monocytogenes*. A full list of accessions can be

**Table 3** Application of ConFindr to the assessment of contamination in published genomes of *L. monocytogenes*, *S. enterica*, and *E. coli*.

| Species | Intraspecies contamination | Cross-species contamination |
| --- | --- | --- |
| *L. monocytogenes* | 27/500 (5.4%) | 5/500 (1%) |
| *S. enterica* | 17/500 (3.4%) | 1/500 (0.2%) |
| *E. coli* | 26/500 (5.2%) | 2/500 (0.4%) |

found in Table S4. Of the 1,500 samples examined, 78 (5.27%) were determined to be contaminated by ConFindr (Table 3, Table S4). For all species, intraspecies contamination appeared to be more prevalent than cross-species contamination. Assembly metrics (number of contigs, N50, genome assembly size) for contaminated samples were compared to samples determined to be clean within each species (Fig. S1). Metrics for contaminated samples were generally within ranges observed for samples determined to be clean. As with the simulated data, coverage levels did not impact intraspecies contamination detection. Of the 100 samples with the highest level of coverage (average coverage depth 176 to 6,148, Table S4), only three samples were deemed to have intraspecies contamination. In a comparator set of 100 samples with a coverage depth ranging from 40 to 50.5, five samples were determined to be contaminated. In the set of isolates with a high coverage level, cross-species contamination was detected in three samples (ERR2135583, ERR1100941, SRR1206152) compared to two samples (ERR2520947, SRR7298866) in the set with lower coverage. It is notable that cross-species contamination would have been predicted for the latter two samples based on poor assembly metrics. For samples with higher coverage, assembly metrics were within typical ranges and detection of contamination may be a result of an increased sensitivity for cross-species contamination at this coverage level.

### Confirmation of contamination in *E. coli* WGS data

In a recent WGS analysis performed in our laboratories, hybrid assemblies of *E. coli* samples led to an incorrect serotype determination (Table 4, sample 1, isolate 3). Three strains of *E. coli* from two samples were sequenced in duplicate or triplicate. Presumptive contamination was identified due to higher number of contigs or larger genome size relative to duplicates from the same sample (Table 4, bold). In one sample, the serotype of an isolate was incorrectly determined (O159:H2). Contamination was confirmed by analysis of samples with ConFindr (Table 4, bold).

### Runtime considerations and installation

ConFindr can be installed with a single command via bioconda (*Grüning et al., 2018*), and completes analysis on a sample in under one minute when using 4 threads and less than 4 GB of RAM. These features make it practical for ConFindr to be installed and run as a standard quality control step in bioinformatics pipelines.

## DISCUSSION

In creating ConFindr, we wanted a tool that would be broadly applicable to the bacterial domain, while also providing enough resolution to detect contamination between closely

**Table 4  Intraspecies contamination in *E. coli*.**

| Sample | Isolate | Serotype | MLST | rMLST | Contigs | N50 | Genome size | ConFindr |
|--------|---------|----------|------|-------|---------|-----|-------------|----------|
| 1 | 1 | O159:H19 | 1611 | 52368 | 76 | 187212 | 5087345 | Clean |
|   | 2 | O159:H19 | 1611 | 52368 | 73 | 187212 | 5085921 | Clean |
|   | 3 | **O159:H2** | 1611 | 52368 | **193** | 186411 | **5500652** | Contaminated |
|   | 4 | O83:H31 | 372 | 1854 | 31 | 436072 | 4967515 | Clean |
|   | 5 | O83:H31 | 372 | 1854 | **112** | 433519 | **5077348** | Contaminated |
| 2 | 1 | O8:H28 | 4496 | 33427 | 55 | 218837 | 4858743 | Clean |
|   | 2 | O8:H28 | 4496 | 33427 | 55 | 218837 | 4857891 | Clean |
|   | 3 | O8:H28 | 4496 | 33427 | **151** | 218837 | **4985955** | Contaminated |

related isolates. We selected the 53 ribosomal protein genes used in the rMLST scheme as they are present in all bacteria and provide enough diversity for high-resolution characterization (*Jolley et al., 2012*). While there are duplicate copies of some of the genes within the scheme (e.g., *L. monocytogenes* has two alleles of BACT000014), the scheme is actively curated, and these exceptions are known and handled by ConFindr. The advantage of this core-gene approach is that the tool can be integrated into pipelines aimed at the analysis of multiple bacterial species. While single-copy core genes have been used to evaluate the quality and completeness of metagenomic assemblies (*Hess et al., 2011*; *Parks et al., 2015*), this approach has not been commonly applied to bacterial isolate WGS data. In the current study, we found ConFindr performed equally well for three species, including both Gram-positive and Gram-negative bacteria, and we would expect similar performance for other species covered by the rMLST database. Due to its reliance on the rMLST scheme, ConFindr is intended for use in bacteria and is not for the detection of contamination in archaeal or eukaryotic samples. Nonetheless, a similar approach of using broadly-conserved core single-copy genes would likely be effective for addressing contamination within other domains.

The sensitivity of ConFindr for detection of intraspecies contamination is dependent on sequence coverage, as well as the number of SNVs occurring within the conserved ribosomal proteins genes used in the analysis. ConFindr is unable to detect contamination if the contaminating isolate has fewer than 3 SNVs within the rMLST genes used in the tool. This cut-off was chosen as ConFindr will occasionally detect one or two false positive cSNVs in rMLST genes (Fig. 1), but we have yet to see an example with 3 or more false positive cSNVs. In practice, this means that ConFindr may sometimes miss contamination between two strains that have only one or two SNVs within the rMLST genes. However, our analysis of the rMLST database demonstrates that greater than 99% of the rMLST profiles for *L. monocytogenes*, *S. enterica* and *E. coli* differed by more than 3 SNVs relative to all other profiles in the database indicating that this tool would generally be effective for detection of contamination with unrelated strains (Fig. 2).

The combined length of the rMLST genes is approximately 20 kilobases, representing only 0.7% of the genome in *L. monocytogenes* and 0.4% of the genome in *E. coli*. Examining this small fraction of the genome limits the sensitivity of ConFindr. This limitation could

be overcome by using core-genome multi-locus sequence typing (cgMLST) schemes for species where they are available; however, doing this would increase the size of the databases used by ConFindr, increase the runtime, and would require additional manual curation of the cgMLST schemes used to ensure reliability in an automated system. Moreover, we found ConFindr to be more sensitive than CheckM for intraspecies contamination, despite the use of a smaller number of core genes relative to CheckM which uses a larger number of lineage-specific core genes. This is likely because ConFindr works at the read level while CheckM works on assemblies (*Parks et al., 2015*). If contamination is at a low level (e.g., one SNV in a gene, at a low contamination percentage), variant positions would likely get lost in the assembly process, limiting the sensitivity of assembly-based approaches. Furthermore, different assemblers or read preprocessing steps may change the results found by assembly-based tools for contamination detection.

In the present study, cross-species contamination was easily identified based on *de novo* assembly metrics (Table S1); however, assemblies with low-level intraspecies contamination had assembly metrics similar to uncontaminated assemblies (Table 2). This is consistent with observations in our laboratory (Table 4). Typical assembly metrics vary among species and strains, making it difficult to develop robust standards for these metrics. For example, *S. enterica* genomes tend to have higher N50 values and assemble into fewer contigs than *E. coli* (e.g., Table 2). Ultimately, this variability makes it difficult to develop standard cutoffs that can be integrated into automated tools.

We applied ConFindr to the evaluation of 1,500 samples in the public SRA repository and identified intraspecies contamination in 5.13% of the samples (Table 3). Notably, intraspecies contamination was more prevalent than cross-species contamination. A recent assessment of 67,758 publically-available *Salmonella* sequences determined that 1.87% of samples had cross-species contamination based on a read classification approach (*Robertson et al., 2018*). Prevalence of cross-species sequence contamination in public repositories is a known issue that has been described in a number of studies (*Merchant, E Wood & Salzberg, 2014*; *Mukherjee et al., 2015*; *Lee et al., 2017*; *Cornet et al., 2018*). Very few studies have looked at intraspecies contamination in public repositories, and we could not identify any studies evaluating prevalence of intraspecies contamination in foodborne pathogens. While the effects of intraspecies contamination are poorly understood, the relatively high proportion of samples determined to be contaminated by ConFindr highlights the need to further investigate the impacts of intraspecies contamination on WGS-based analyses.

WGS pipelines for public-health microbiology often include analyses of SNVs among a group of isolates to assess evolutionary relatedness and/or detection of genetic targets (e.g., serotype markers, virulence determinants) (*Lambert et al., 2015*; *Ronholm et al., 2016*; *Allard et al., 2016*; *Chen et al., 2017*). The impact of using contaminated data in these analyses is not well understood as validation schemes for bioinformatics pipelines do not often assess the effect of contamination on results of analyses to determine acceptable limits. We found one report on the development and validation of the SNVPhyl pipeline that incorporated an assessment of the impact of contamination (*Petkau et al., 2017*). In this evaluation, the number of SNVs detected decreased as contamination with a closely related strain increased, and detection of clusters of epidemiologically-related isolates was

impacted with greater than 10% contamination (*Petkau et al., 2017*). While few studies of the impact of contamination on phylogenomic analyses exist, most SNV detection pipelines use cut-offs for coverage and relative nucleotide abundance at a given position to ensure validity of a SNV (*Davis et al., 2015*; *Petkau et al., 2017*). The presence of intraspecies contamination in the analysis of a sample would ultimately result in the exclusion of valid SNVs and could have impacts on the resulting phylogenetic tree topology. Contamination may have more important effects on detection of genetic markers in WGS data. For example, in a recent analysis in our laboratory, contamination impacted the accuracy of the determination of an *E. coli* serotype (Table 4). Similarly, intraspecies contamination could result in detection of virulence and antibiotic resistance genes, as well as pathogenicity islands or other horizontally acquired genes that are part of the contaminant strain and not the target strain.

## CONCLUSION

We have developed a novel bioinformatics pipeline (ConFindr) for detection of contaminating reads in raw short-read bacterial WGS data and have demonstrated its applicability for quality assessment of data derived from the priority foodborne pathogens *L. monocytogenes*, *S. enterica* and STEC. To our knowledge, this is the first automated tool developed specifically for this purpose. ConFindr outperforms existing bioinformatics tools for detection of intraspecies contamination in bacterial WGS data and can reliably detect cross-species contamination. It may be possible to adapt the approach used by ConFindr for long read data, but this is reserved for future work due to the drastic differences in error profile that may impact the identification of contaminating SNVs. ConFindr should be universally applicable to bacterial genomes and can be easily implemented in quality-control pipelines for WGS analysis. Further studies are needed to better understand the effects of contamination on WGS analyses, as well as establish what acceptable levels of contamination are when analyzing WGS data. The integration of tools such as ConFindr in quality-analysis pipelines will improve the reliability of public WGS databases, and the accuracy of downstream analyses.

ConFindr source code and documentation are freely available under the MIT Licence at https://github.com/OLC-Bioinformatics/ConFindr.

## ACKNOWLEDGEMENTS

We gratefully acknowledge Julie Shay, Mohamed Elmufti and Austin Markell for helpful comments on the draft manuscript. We would also like to thank Katie Eloranta and Jennifer Liu for providing sequence data used in Table 4, and users who have tested ConFindr and suggested improvements and modifications.

### Funding

This work was funded by the Canadian Food Inspection Agency (OLC-F-1803). The funders had no role in study design, data collection and analysis, decision to publish, or preparation of the manuscript.

### Grant Disclosures

The following grant information was disclosed by the authors:
Canadian Food Inspection Agency: OLC-F-1803.

### Competing Interests

The authors declare there are no competing interests.

### Author Contributions

- Andrew J. Low conceived and designed the experiments, performed the experiments, analyzed the data, prepared figures and/or tables, authored or reviewed drafts of the paper, approved the final draft.
- Adam G. Koziol conceived and designed the experiments, contributed reagents/materials/analysis tools, authored or reviewed drafts of the paper, approved the final draft.
- Paul A. Manninger performed the experiments, approved the final draft.
- Burton W. Blais conceived and designed the experiments, contributed reagents/materials/analysis tools, authored or reviewed drafts of the paper, approved the final draft.
- Catherine D. Carrillo conceived and designed the experiments, analyzed the data, contributed reagents/materials/analysis tools, prepared figures and/or tables, authored or reviewed drafts of the paper, approved the final draft.

### DNA Deposition

The following information was supplied regarding the deposition of DNA sequences:
Raw data are available at DDBJ/EMBL/GenBank under BioProject PRJNA507762. The accession numbers and strain descriptions are listed in Table S1.

### Data Availability

Raw WGS data are available at DDBJ/EMBL/GenBank under BioProject PRJNA507762.
The accession numbers and strain descriptions are listed in Table S1.
https://www.ncbi.nlm.nih.gov/bioproject/PRJNA507762.
Code for ConFindr is freely available at
https://github.com/OLC-Bioinformatics/ConFindr.

### Supplemental Information

Supplemental information for this article can be found online at http://dx.doi.org/10.7717/peerj.6995#supplemental-information.

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
