# Peer review of "ConFindr: rapid detection of intraspecies and cross-species contamination in bacterial whole-genome sequence data"

_PeerJ, doi:10.7717/peerj.6995_

## Round 0.1 · original submission · Minor Revisions

Please answer all reviewer's questions and/or comments in your revised version. One thing to note is that although they're not for bacterial genomes, there are multiple tools that are commonly used to detect intra-species contamination for human genome sequences. It'd be helpful if you could discuss how these methods might and might not be related to the proposed method.

·

Basic reporting

Specific comments

1) Figure 1 Legend. Missing closing parenthesis on “(Supplementary Table S3”.

Experimental design

Specific comments

1) Coverage – The coverage evaluated in table S1 ranged from approximately 30X to 150X. Coverage can frequently exceed 300X-500X for isolates sequenced on a HiSeq or high-priority isolates that need to be quickly run on a MiSeq with a small number of samples. It would be helpful to evaluate the effect of coverage on uncontaminated isolates to ensure that sequencing artifacts don’t appear as contamination.
2) Line 103 – Parameter settings for ART are not provided in the paper or in the supplementary material.
3) The authors used version 3 chemistry at 2x300 sequencing conditions to generate their real data and also their simulated data (see point 2 above, simulation conditions not provided). Many public health laboratories are using 2x150 or 2x250 chemistry. The authors should evaluate the performance of their software with shorter reads. Note – if the simulation conditions are provided then it should not be necessary to submit the synthetic data to NCBI.

Validity of the findings

Specific comments

1) Table S4. It would be helpful to include BioSample accessions along with metrics such as number of reads and skesa output (coverage depth, number of contigs, assembly size) for the SRA samples. Some of these appear to have very high coverage and it’s not clear how this may influence the ConFindr results.
2) Lines 189-191. There are 1500 SRA accessions in table S4 but line 189 states 500 isolates. Is this a typo or do the 1500 SRA accessions include replicates?
3) Figure 1. Standard error bars are shown but the number of replicates for each bin is not reported.

Additional comments

The paper is clear, well organized, and the software addresses a pressing need for assessment of contamination in multiplexed short read sequencing runs in public health laboratories. The source code appears to be well written and the software documentation is complete. Lastly, the software installed easily using BioConda.

ConFindr is available under the MIT license but the rMLST data base required to run the software is under a much more restrictive license (https://pubmlst.org/rmlst/rMLST_licence.pdf). Many public health laboratories may not qualify for the pubmlst academic use license. I strongly encourage the authors to distribute the software with a stripped-down version of an rMLST database compatible with the MIT license containing public sequences for Salmonella, Listeria, STECs, and other priority pathogens. The bulk of the contamination events identified in SRA in the supplementary material appear to be intraspecies so having a smaller database with fewer species wouldn’t seem to have too much of an impact.

Unfortunately I’m not able to test the current version of ConFindr, 0.5.1, as my pubMLST request for rMLST access is still pending. I have tested the 0.4.3 version of the software and I can confirm that it produced results consistent with other methods for evaluating contamination.

Reviewer 2 ·

Basic reporting

The manuscript is clearly and professionally written, with attention to detail in text, figures and table. A good background is provided with relevant references throughout.

Experimental design

The manuscript clearly states the impetus for the work and distinguishes it from existing tools. Rigorous experiments are performed to validate the work and methods are well described, with source code available and datasets submitted to public repositories. The one major sticking point in availability is that, for the software to work, it seems necessary to email a single person named Keith and await a response granting access to a database. If this tool is to be widely incorporated into pipelines, as suggested, this does not seem practical or sustainable. Perhaps a static version of the database could be provided for quick start-up, with the license-restricted update left as an option?

Validity of the findings

The findings are for the most part well-supported. The one area where I felt that the claims went slightly further than the experiments showed was the superior sensitivity versus existing tools, Kraken and CheckM. These tools were given rather arbitrary cutoffs for positive identification. While the 0.5% cutoff for Kraken seems a lot lower than the 5% ConFindr requires for each SNV, these are not necessarily comparable. Since Kraken is a genome-wide classifier, many reads will simply be uninformative for closely-related strains, but a few classified to strain level could still be meaningful for finding contamination. Though CheckM is an MLST based method like ConFindr, its contamination percentages may simply be skewed by assembly rather than missing the contamination completely. Rather than anecdotal experience for needing to investigate CheckM's results, I would like to see some kind of analysis of false positives to justify the cutoffs for both CheckM and Kraken. A complete ROC may not be necessary, but perhaps at least loosen the thresholds until these tools are as sensitive as ConFindr and see whether false positives do indeed occur.

Additional comments

The manuscript is thoughtful, well-motivated, and clear. Though the central claim of improved sensitivity could use bolstering as described above, the manuscript in general benefits from a wide variety of well-documented experiments, and the generation and inclusion of real instrument data is appreciated.

---

## Round 0.2 · accepted · Accept

Please note that PeerJ does not offer proofreading or editorial services so please make sure that you do not have typos or figure formatting errors while in production.

# ·

Basic reporting

no comment

Experimental design

no comment

Validity of the findings

no comment

Additional comments

The revisions address the concerns I noted in my first review. Unfortunately the authors were not able to resolve the database license issue for this release of their tool. Hopefully future versions of ConFindr will include a static database that is unencumbered by the pubmlst rMLST license.

Reviewer 2 ·

Basic reporting

The revisions are written to the same high quality as the original submission.

Experimental design

The authors address the database licensing issues with plans to update the distribution with priority-specific databases in the future.

Validity of the findings

The authors have performed and described additional experiments showing that false positives occur when attempting to make CheckM and Kraken more sensitive, and that ConFindr's sensitivity and specificity are indeed superior.

Additional comments

Concerns in the original manuscript have been thoughtfully and thoroughly addressed.